# Lung deflation while placing a subclavian vein catheter: Our experience in minimizing the risk of pneumothorax

Daher K. Rabadi[1]*, Ahmad K. Abubaker[1], Sami A. Almasarweh[2]

1 Faculty of Medicine, Department of Anesthesiology, Jordan University of Science & Technology, Irbid, Jordan, 2 Faculty of Medicine, Jordan University of Science & Technology, Irbid, Jordan

* daherrabadi@yahoo.com.au

**Data Availability Statement:** The dataset is not available publicly according to Institutional Review Board (IRB) decision (decision reference number: 15-2011) however it can accessed after obtaining the approval of the ethics committee at King

## Abstract

### Purpose

Lung deflation may reduce the risk of pneumothorax based on the assumption that the distance between the subclavian vein and the lung pleura would increase as well as the diameter of the vein. We aim to provide evidence to support the suggested desideratum of deflation in adults.

### Methods

A prospective database was created that included patients who underwent subclavian vein catheterization for monitoring and therapeutic reasons from January 2014 to January 2020. Measurements using ultrasonography of the diameter of the subclavian vein were taken while the patient's breathing was controlled by a ventilator and then repeated after disconnecting the mechanical ventilation and opening the pressure relief valve.

### Results

A total of 123 patients were enrolled, with an average age of 41.9 years. The subclavian vein diameter was measured during controlled breathing with a mean average of 8.1 ± 0.6mm in males and 7.1 ± 0.5mm in females. The average increase after lung deflation with the pressure relief valve closed was 8.0± 5.1mm in males and 13.9 ± 5.4mm in females. An increase was noticed after opening a pressure valve, and the means were 5.5 ± 2.8mm in males and 5.1 ± 3.3mm in females. The catheter malposition rate was 0.8%

### Conclusion

The benefit of interrupting mechanical ventilation and lung deflation lies within possibly avoiding pneumothorax as a complication of subclavian vein catheterization. These findings support the need for evidence regarding the curtailment of pneumothorax incidence in spontaneously breathing patients and the suggested increase in first-time punctures and success rates.

Abdullah University Hospital and Jordan University of Science & Technology by contacting the following email: IRB@just.edu.jo.

**Funding:** The author(s) received no specific funding for this work.

**Competing interests:** The authors have declared that no competing interests exist.

## Introduction

Central venous catheters are imperative in the modern era of medical practice. Defined as a catheter insertion in a great venous vessel, commonly placed in the internal jugular vein or subclavian vein to have the catheter end in the superior or inferior vena cava or the right atrium [1,2]. Nearly 8% of hospitalized patients have a central venous catheter and access site [3].

The Seldinger technique exhibits precedence in the choice of physicians. To mitigate the risk of complications, the utilization of ultrasound guidance in visualizing the vessel has been proven to increase the success rate of the aforementioned procedure [4]. Cannulating the subclavian vein is part of common medical practice and offers particular advantages when compared to other access sites. In comparison to other venous access points, patients who had subclavian catheters experienced fewer infectious complications [3,5]. Moreover, the access provided allows for the risk of a thrombotic event to be lower compared to other vein accesses and the ability to sustain patency in hypovolemic patients to be improved, along with better patient comfort and nursing care [6–8].

The advantages of the subclavian vein catheters is being more accessible to the operator in trauma patients, and it can be placed without disrupting the airway management during the initiatory phases of resuscitation. Failure rates vary depending on the physician's experience, patient's conditions, and clinical settings; nonetheless, complications can be avoided using proper techniques [9,10]. Prompt complications of such procedure include subclavian artery puncture, pneumothorax, hemothorax, chylothorax, mediastinal hematoma, and possible injury to the phrenic nerve [11,12].

The use of the subclavian vein may have its benefits compared to other large veins; however, the possibility of leading to pneumothorax is significant. The distance between the subclavian vein and the lung pleura is vital in reducing this possibility [13]. The proximity of the lung pleura to the vein varies with breathing. During full expiration, the lung volume will be decreased, maximizing the distance between the vein and pleura, thereby, in theory reducing the risk of pneumothorax. Furthermore, the effect of inspiration on the collapsibility of the vein can negatively affect the success rate of the catheter placement. The increase of the intrathoracic pressure during full expiration and the lung deflation can increase blood volume in the subclavian vein, making it more accessible when combined with the increase in space between the pleural and the aforesaid vein [14]. Mechanical ventilation is halted to prevent injury to adjacent structures during the procedure and prevent procedural lung injury [15]. Clinicians have used such a technique at their own discretion without clear evidence regarding its benefit. Despite the fact that this may increase the safety margin of the procedure, deflating the lung may cause hypoxemia in patients, and the occurrence of pneumothorax would not be wholly eliminated [7,16].

In this study, we aim to study the effect of lung deflation in previously spontaneously breathing adults who are being ventilated through intermittent mandatory ventilation during subclavian vein catheterization with respect to the diameter of the subclavian vein. This study also takes into consideration the rate of complications concerning apnea induced by disconnecting the ventilation machine and the catheterization procedure.

## Methodology

A database was prospectively that included patients who underwent subclavian catheterization for monitoring and therapeutic causes between January 2014 and January 2020 at King Abdullah University Hospital (KAUH), a tertiary hospital affiliated with the Jordan University of Science and Technology (JUST). The database included adult patients that were operated on electively, excluding patients with infections at the puncture site, pregnant women, children, and patients who have congenital malformations. The patients enrolled had a central line

placed in the subclavian vein which was performed by the most senior anesthesia consultant at KAUH, located in the north of Jordan. All the patients consented verbally and signed a written consent witnessed by the senior anesthesia consultant, escorting close family relative and the senior responsible nurse according to KAUH ethical protocol.

## Approach

All the patients who were scheduled for surgery were instructed to follow the fasting protocol according to their care team. Upon arrival to the operating suite, a noninvasive arterial blood pressure monitor, and a pulse oximeter are attached for monitoring, and an electrocardiogram is done. After general anesthesia is induced, mechanical ventilation is initiated, and the respiratory rate is adjusted to maintain end-tidal CO2 of approximately 40mmHg. The anesthesia machine used was GE Datex Ohmeda Avance S5 (Datex-Ohmeda, Ohmeda Drive, Madison, Wisconsin, USA). An initial tidal volume of 6–8 ml/kg is used, and the positive end-expiratory pressure is calculated based on the patient's bodyweight and set in mbar.

The patients were placed supine on a horizontal table with their heads in the midline position with arms adducted next to them. The arm corresponding to the site of the procedure is extended at the shoulder level to increase the distance between the clavicle and subclavian vein. After proper sterilization and draping of the area, including the neck and chest above the nipple line. 100% oxygen flow with inhaled anesthetic is used to maintain anesthesia and pre-oxygenation of the patient. After 1 minute, the patients were safely placed in the Trendelenburg position. The ultrasound probe would be placed parallel to the clavicle superiorly with slight acute angulation towards the sternum, offering a coronal view (Fig 1).

If the patient is a female, an assistant will apply pressure to the right breast tissue pulling it caudally to avoid having the tissue infiltrate the sterile field. The positive end-expiratory pressure is then stopped inducing apnea, and after 3 seconds, the ultrasound probe is placed in the same position previously mentioned. The patient is then reoxygenated for a short moment, and then with mechanical ventilation stopped, we opened the pressure relief valve, causing equivalency between the pressure within the pleural compartment and atmospheric pressure.

The ultrasound probe would remain at the same place during the intervention. The measurements were recorded manually and noted in a database without recording any personal or identifying details of the patients. The puncture was usually performed on the right side unless when there were any contraindications such as the presence of port catheters, axillary dissections, or ipsilateral shunts). Under sterile conditions, the puncture was performed using the Selinger technique. A 20cm long three-lumen catheter with a 0.32-inch diameter spring-wire guide is soft on one end and a "J" tip on the other (Arrow International LLC, Morrisville, North Carolina, USA). The needle was inserted at the junction between the lateral third of the length of the clavicle and the medial two-thirds. Correct catheter position was verified by ultrasound and steady flow of dark blood with pressure transduction.

## Postprocedural protocol

Post-procedural complications were defined as pneumothorax, hemothorax, arterial puncture, missed puncture, subcutaneous hematoma, site infection, catheter malposition, chylothorax, air embolism, cardiac arrest. In addition, a postoperative chest x-ray was performed between 6 and 8 hours to rule out pneumothorax.

## Statistical analysis

A paired t-test was used to compare the mean difference of subclavian vein diameter after every step describe in the manuever. The statistical analysis was two-tailed, and the

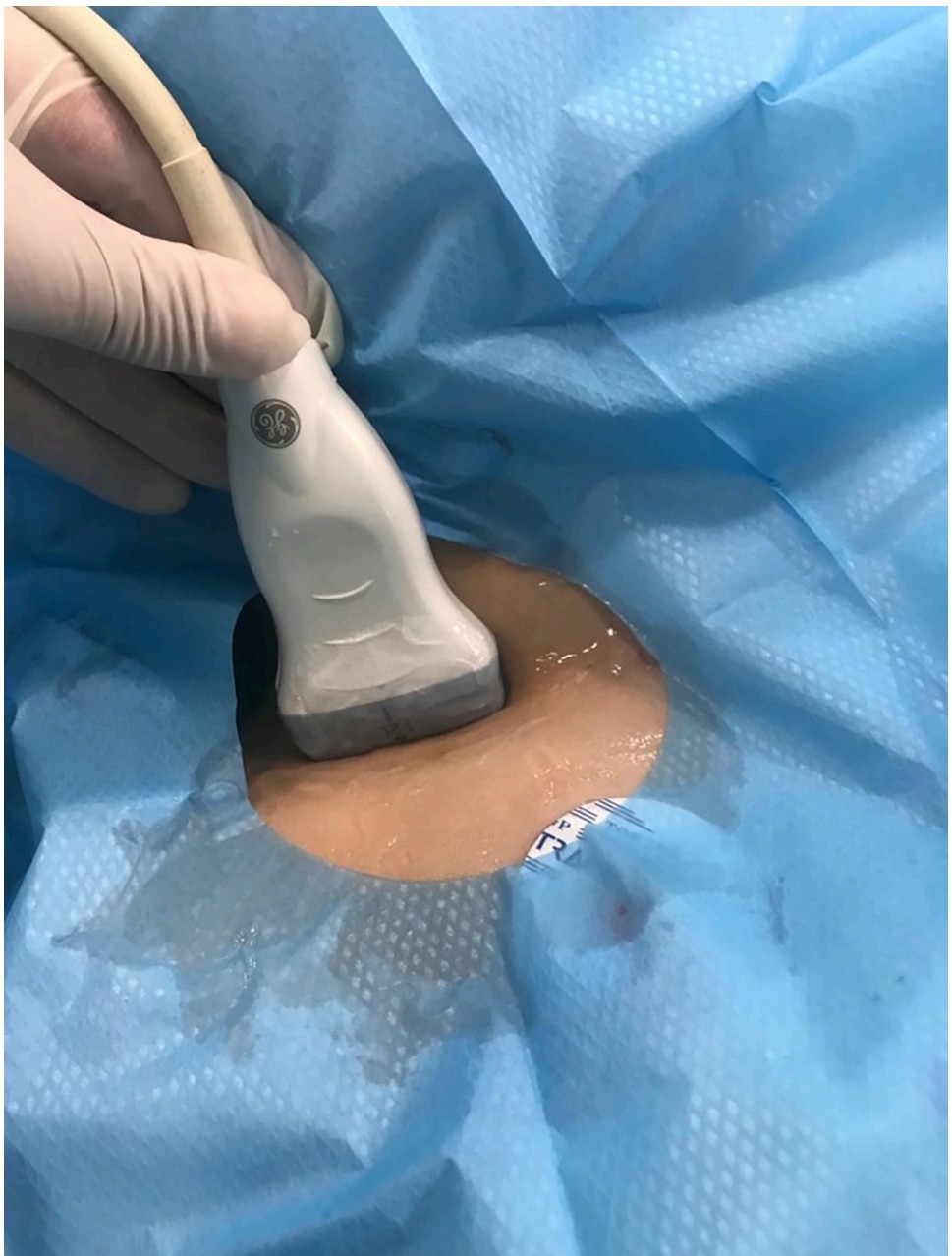

**Fig 1. Position of ultrasound probe on the patient.**

significance threshold was set at 0.05 or less. The IBM® SPSS® Statistics version 26 (IBM, Armonk, New York, United States) was used for statistical analysis, and the institutional review board approved the study before the data collection commenced.

## Ethical statement

The Institutional Research Board (IRB) at King Abdullah University Hospital and Jordan University of Science and Technology approved this study (decision reference number: 15–2011). Data collected was kept strictly confidential and was only analyzed for the purpose of this

study. All the patient have signed a consent form as part of standard procedures at King Abdullah University Hospital and Jordan University of Science and Technology.

## Results

A total of 123 patients were enrolled in the database of patients who had elective surgery, a subclavian vein catheter placed by the same physician, and fit our inclusion criteria. The mean of the ages was 42.6 years old, with the lowest being 18 years old and the highest being 93 years old. The population had a male predominance as 54.5% of the patients were identified as males (67 patients) and 45.5% as females (56 patients). 23.6% of the population were patients who were older than 60 years old. The average body mass indexes for males and females were 26.1 ± 4.0, and 24 ± 3.8 kg/m², respectively. Table 1: Demographic Data.

The subclavian vein diameter during controlled breathing was measured in males with a mean average of 8.1 ± 0.6mm and 7.1 ± 0.5mm in females. The exact measurements were repeated after stopping the ventilation but without opening the pressure relief valve with an average of 8.9 ± 0.3mm and 8.5 ± 0.4mm in males and females, respectively. After opening the pressure relief valve, we observed a further increase in the diameter as the measurements were repeated while maintaining the same probe position in all patients; males and females were 9.4 ± 0.3mm and 8.9 ± 0.2mm, respectively. (Figs 2 and 3) There was no correlation between the increase in age and the changes in the diameter of the subclavian vein regardless of the state of the pressure valve. We noticed a statistically significant difference between the changes in diameter in males and females after disconnecting the mechanical ventilations and before opening the pressure relief valve. The mean change in females was 13.9 ± 5.4mm and 8.0 ± 5.1mm in males, with a p-value of 0.001(95% CI 12.44%-15.34%) and 0.001(95%CI 6.67%-9.24%) respectively.

The catheter malposition rate was 0.8% (1 patient), and no arterial punctures were observed. All the patients were attentively monitored for significant complications. There was no incidence of hemothorax or pneumothorax during the operations; however, one patient developed Pneumothorax four days postoperatively, and it was resolved immediately. The oxygen saturation level dropped momentarily below 95% in three patients (2.43%) during deflation who have been adequately ventilated afterward without any complications. All the patients were successfully weaned off of mechanical ventilation, and the average duration of hospital stay was five days.

## Discussion

The risk of pneumothorax is substantially higher in subclavian venous catheterizations compared to the other sites, and it is one of the most common mechanical complications of the

**Table 1. Patient demographic data.**

|  | Number of Patient n (%) |  |
| --- | --- | --- |
| Total Number of Patients | 123 (100%) |  |
| Male | 67 (54.5%) |  |
| Female | 56 (45.5%) |  |
| Older than 60 years of age | 29 (23.6%) |  |
| Measure | Mean | Standard deviation |
| Age | 42.6 | 19.8 |
| Body Mass Index |  |  |
| Male | 26.1 | 4 |
| Female | 24 | 3.8 |

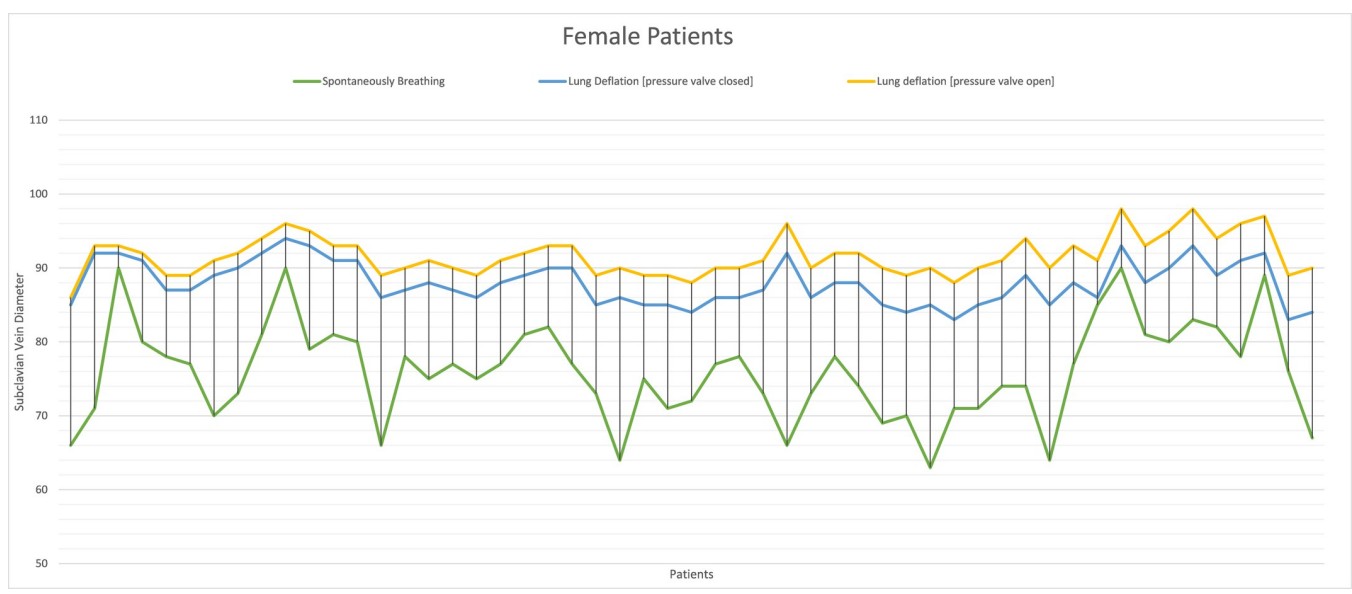

**Fig 2. Changes in diameter of the subclavian vein in female patients.** The green line represents the diameter of the subclavian vein when the patient is spontaneously breathing, the blue represents the diameter after causing lung deflation with pressure relief valve close and the yellow line is after opening the pressure relief valve. Each vertical line on the plot graph represents the diameter of the subclavian vein for each female patient therefore the distance between the cross-section point of the vertical line and the horizontal lines represents the difference in diameter per patient.

procedure.(6) It has been suggested that lung deflation may reduce the risk of pneumothorax based on the assumption that the distance between the subclavian vein and lung pleura would increase. However, there was an apparent lack of evidence to support the suggested desideratum of deflation in adults.

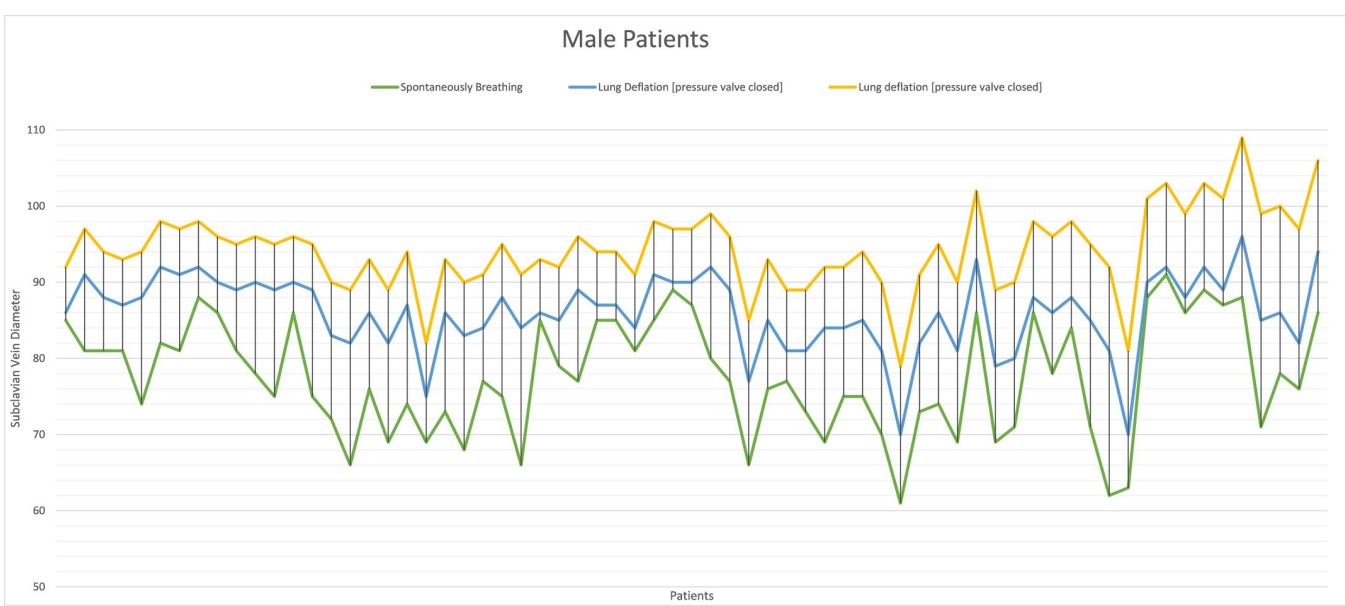

**Fig 3. Changes in diameter of the subclavian vein in male patients.** The green line represents the diameter of the subclavian vein when the patient is spontaneously breathing, the blue represents the diameter after causing lung deflation with pressure relief valve close and the yellow line is after opening the pressure relief valve. Each vertical line on the plot graph represents the diameter of the subclavian vein for each male patient therefore the distance between the cross-section point of the vertical line and the horizontal lines represents the difference in diameter per patient.

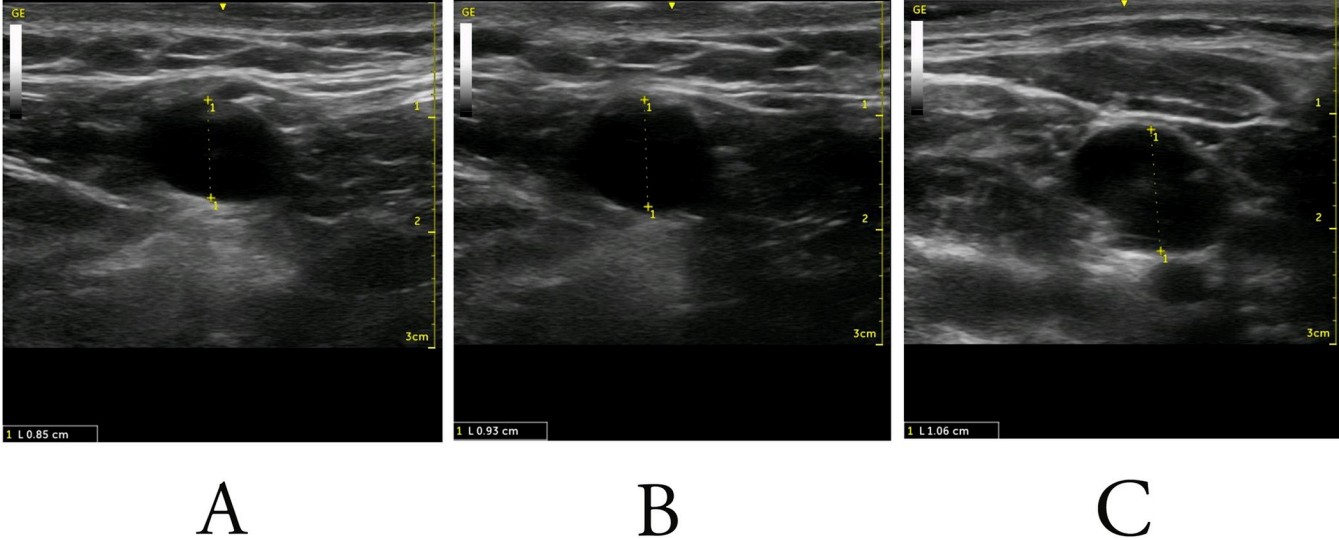

**Fig 4. A- Ultrasound Image showing the subclavian vein diameter while the patient is mechanically ventilated.** B- the same plane showing the subclavian vein diameter while the patient is off mechanical ventilated and lung deflation is initiated. C- the subclavian vein diameter after opening the pressure release valve.

In this study, an increase was observed in the diameter of the subclavian vein in a considerable majority of our patients. (Fig 4) Lim et al. and Hightower et al., reported a similar increase in the subclavian vein cross-sectional area; however, their results were not statically significant [14,17]. Relating our findings to the risk of complications, the risk of pneumothorax compared to the literature is significantly lower [5,18,19]. Furthermore, previous studies evaluating mechanical ventilation and lung deflation did not perform the procedure under ultrasound guidance which has been proven to increase success rates compared to landmark-guided methods [13,16]. A study comparing continued and interrupted mechanical ventilation during the procedure confirms a low pneumothorax incidence when inducing apnea [20]. However, the aforementioned study performed the venipuncture with the pressure relief valve closed. In this study, the pressure relief valve was briefly opened after the end-expiration period and interruption of mechanical ventilation. Opening the valve allowed for the equivalency between the intrathoracic and atmospheric pressure to occur and increase the lung deflation, thereby increasing the diameter of the subclavian vein.

This study has not previously highlighted an added benefit to lung deflation. Compared to spontaneously breathing patients, lung deflation prevented the movement of organs within the thoracic cavity and chest wall. This facilitates having a more stable reference line while performing this procedure and reduces the risk of penetrating injury to adjacent structures. In addition, ventilation was resumed after the guidewire was securely inserted, not after placing the catheter. This slight alteration was implemented to ensure patients' safety and avoid hypoxia.

## Limitations

This study has its strengths and limitations. All the patients in our database had a subclavian vein placed by the same senior anesthetist who has placed over countless central lines in his career. This is to make sure that our complications and success rates are not affected by the lack of experience of the practitioner performing the procedure. The percentages of male and female patients are also relatively similar as there are gender-related anatomical differences or

predisposing risk factors. One of the limitations of this study is the quality of plane ultrasound images provided and the lack of experience of the responsible team in the surgical suite using the ultrasonography machine for subclavian vein catheterization. This would negatively affect the possibility of standardizing the measurement procedure for the subclavian vein caliber as the operators hands would move during canulation. All the venipunctures were approached in the infraclavicular plane, and the supraclavicular approach was not considered. The database was created after the data collection phase ended, and it excluded patients that were critically ill and were exposed to high PEEP ventilation. Another limitation would be that the majority of the patients did not repeat the chest x-ray when discharged, and the possibility of detecting late-onset pneumothorax was not possible. The sample size is relatively small and increases the possibility of type II error during our analysis of our findings. Our study and findings warrant further assessments of apnea induction during central line placement in the subclavian vein, especially in critically ill patients. Further developments in evidence add to the technique's superiority in terms of success rates and pneumothorax incidence and thereby promote the installation of guidelines ensuring patient safety and well-being. This is study is also highlighting the importance of standardizing vascular access protocols to include ultrasound guidance with advanced procedural maneuvers to improve its effectiveness in terms of clinical outcomes [21,22].

## Conclusion

The benefit of interrupting mechanical ventilation and lung deflation lies within conceivably avoiding pneumothorax as a complication of subclavian vein catheterization. The findings within this study support the need for evidence regarding support the curtailment of pneumothorax incidence and the suggested increase in first-time punctures and success rates in healthy, spontaneously breathing patients.

## Author Contributions

**Conceptualization:** Daher K. Rabadi, Sami A. Almasarweh.

**Data curation:** Ahmad K. Abubaker, Sami A. Almasarweh.

**Formal analysis:** Sami A. Almasarweh.

**Investigation:** Daher K. Rabadi.

**Methodology:** Daher K. Rabadi, Sami A. Almasarweh.

**Project administration:** Daher K. Rabadi, Ahmad K. Abubaker.

**Supervision:** Daher K. Rabadi, Ahmad K. Abubaker.

**Validation:** Daher K. Rabadi, Ahmad K. Abubaker, Sami A. Almasarweh.

**Writing – original draft:** Sami A. Almasarweh.

**Writing – review & editing:** Daher K. Rabadi, Ahmad K. Abubaker.

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
