## [Decision Letter · Decision Letter 0]

1 Sep 2022

PONE-D-22-12796Lung Deflation while Placing a Subclavian Vein Catheter: Our Experience in Minimizing the Risk of Pneumothorax.PLOS ONE

Dear Dr. Rabadi,

Thank you for submitting your manuscript to PLOS ONE. After careful consideration, we feel that it has merit but does not fully meet PLOS ONE’s publication criteria as it currently stands. Therefore, we invite you to submit a revised version of the manuscript that addresses the points raised during the review process.

Two external reviewers have evaluated your submission, and have identified a number of concerns that need to be carefully addressed in a revision of the manuscript. Please pay particular attention to Reviewer 2's requests for important methodological clarifications to enable a more thorough evaluation of your findings.

We look forward to receiving your revised manuscript.

Kind regards,

Jamie Males

Editorial Office

PLOS ONE

Journal Requirements:

2. Please provide additional details regarding participant consent. In the ethics statement in the Methods and online submission information, please ensure that you have specified what type you obtained (for instance, written or verbal, and if verbal, how it was documented and witnessed). If your study included minors, state whether you obtained consent from parents or guardians. If the need for consent was waived by the ethics committee, please include this information

In your cover letter, please note whether your blot/gel image data are in Supporting Information or posted at a public data repository, provide the repository URL if relevant, and provide specific details as to which raw blot/gel images, if any, are not available. Email us at plosone@plos.org if you have any questions

Reviewers' comments:

Reviewer's Responses to Questions

**Comments to the Author**

1. Is the manuscript technically sound, and do the data support the conclusions?

Reviewer #1: Yes

Reviewer #2: No

2. Has the statistical analysis been performed appropriately and rigorously? 

Reviewer #1: N/A

Reviewer #2: No

3. Have the authors made all data underlying the findings in their manuscript fully available?

Reviewer #1: Yes

Reviewer #2: Yes

4. Is the manuscript presented in an intelligible fashion and written in standard English?

Reviewer #1: Yes

Reviewer #2: Yes

5. Review Comments to the Author

Reviewer #1: An interesting manuscript in its field, I have no comments. This manuscript contains an interesting subject on the clinical field that any clinical doctor might encounter in the everyday clinical practice.

Reviewer #2: Overall, I think this is an interesting idea and potentially something to consider when placing SC lines. My main concern is that I am not convinced the measurements of the diameter were done accurately and consistently enough, and thus I am not sure of the validity of the results. I think it would be challenging for anyone to consistently measure the diameter of the vein in the same spot every time. I think a better description of how exactly these measurements were done and much better pictures would go a long way towards alleviating this concern. Additional specific comments and suggestions are given below.

Introduction

In general, I think there is a lot of superfluous information in the introduction that could be eliminated (such as the entire first paragraph). I would trim it down and just talk about why subclavian placement is done (advantages of this site over others) and then the purpose of your study.

here is complete unanimity regarding the value of central venous catheters in critically ill

patients. – I am not sure this statement is accurate and it is probably not needed. Consider modifying or deleting.

“Canulating the subclavian vein is part of everyday medical practice…” – I might be a relatively common procedure but I would not say it is part of everyday practice per se.

In terms of allowing for thrombotic events to be “abated” – perhaps it would be more accurate to say it lowers this risk as compared with the IJ or femoral sites

Rather than say the SC is preferred you might want to say “advantages of the SC site over the other options include…”

You might want to change the word fomenting to causing or something similar

Methodology

After 1 minute, the patients were safely placed in the Trendelenburg position. The

ultrasound probe would be placed parallel to the clavicle superiorly with slight acute angulation

towards the sternum, offering a coronal view – Could you include some images ?

It sounds like 1 sole operator performed all the procedures. Is that accurate?

Results

How was the SC vein diameter measured? Short or long axis? At the point of maximal diameter? Can you show an image with measurement ?

Figure 1 and 2 – please label x and y axis and please consider using different colors rather than different subtle shades of blue

What kind of ventilator was used?

Figure 3 is not very clear. The images of the vein look somewhat different in each view and it isn’t clear to me that diameter is really different because it looks like the measurement was not taken at exactly the same spot?

I think the main limitations concern how exactly the measurements were done, and possible lack of confidence that they were done the same way every time. It seems like it would be very easy to obtain different and conflicting results depending on how the measurements were done.

6. PLOS authors have the option to publish the peer review history of their article (what does this mean?). If published, this will include your full peer review and any attached files.

Reviewer #1: No

Reviewer #2: No

---

## [Author Response · Author response to Decision Letter 0]

19 Oct 2022

We are very grateful to you for providing us with the opportunity to submit a revised draft of our manuscript titled, “Title: Lung Deflation while Placing a Subclavian Vein Catheter: Our Experience in Minimizing the Risk of Pneumothorax” to Plos One. We are highly appreciative of the time and effort that have been dedicated to providing us with your valuable feedback on the manuscript. We are also thankful to the reviewers for their insightful comments. We have been able to incorporate changes to reflect most of the suggestions provided and have highlighted the changes accordingly in the manuscript. 

Enclosed herewith is a point-by-point response to the reviewers’ comments and concerns. 

Reviewer #1: 

"An interesting manuscript in its field, I have no comments. This manuscript contains an interesting subject on the clinical field that any clinical doctor might encounter in the everyday clinical practice." 

We are very grateful. The prospects of improving patient’s quality of care depend on repeated assessments of the measures taken. Therefore, this motivates us to explore unproven methods and validate the necessity to always question way to improve patients safety and outcome of care 

Reviewer #2 

"Overall, I think this is an interesting idea and potentially something to consider when placing SC lines. My main concern is that I am not convinced the measurements of the diameter were done accurately and consistently enough, and thus I am not sure of the validity of the results. I think it would be challenging for anyone to consistently measure the diameter of the vein in the same spot every time. I think a better description of how exactly these measurements were done and much better pictures would go a long way towards alleviating this concern. Additional specific comments and suggestions are given below."

Thank you very much for such an insightful comment. We do agree about the limitations caused due to the possible variability in measuring the diameter of the subclavian vein. To minimize the chance of error, We have chosen to have one operator for the whole patient group as the probe pressure and placement might differ from one person to the other. As enclosed in the supporting file attached in the submission, We have addressed the issue by showing one of the initial steps to defining the diameter used in our database which would be taking multiple measurements and choosing the greatest diameter. We believe that a single measurement used can account for the variability in ultrasound probe pressure compared to calculating cross sectional area. We do also agree about the lack of quality of imaging however we were unable to resolve the issue due to lack of modern ultrasound machines in the operating suite.

"In general, I think there is a lot of superfluous information in the introduction that could be eliminated (such as the entire first paragraph). I would trim it down and just talk about why subclavian placement is done (advantages of this site over others) and then the purpose of your study.

here is complete unanimity regarding the value of central venous catheters in critically ill

patients. – I am not sure this statement is accurate and it is probably not needed. Consider modifying or deleting.

“Canulating the subclavian vein is part of everyday medical practice…” – I might be a relatively common procedure but I would not say it is part of everyday practice per se.

In terms of allowing for thrombotic events to be “abated” – perhaps it would be more accurate to say it lowers this risk as compared with the IJ or femoral sites

Rather than say the SC is preferred you might want to say “advantages of the SC site over the other options include…”

You might want to change the word fomenting to causing or something similar" 

"We highly appreciate your input. Accordingly we have made some changes to the introduction to make it shorter and with simpler, less sophisticated vocabulary. 

After 1 minute, the patients were safely placed in the Trendelenburg position. The ultrasound probe would be placed parallel to the clavicle superiorly with slight acute angulation

towards the sternum, offering a coronal view – Could you include some images ?"

Thank you so much. 

We have added a photo of a patient showing the ultrasound probe position against the patient’s clavicle as described 

The photo has been labelled as “fig 1”

"it sounds like 1 sole operator performed all the procedures. Is that accurate?"

That is correct. The most senior anesthesia consultant performed all measurements and venipunctures on the patients. That was done to minimize the limitations due to lack of way to standardize the procedure in terms of complication rates, mispunctures and ultrasound placement

How was the SC vein diameter measured? Short or long axis? At the point of maximal diameter? Can you show an image with measurement 

Multiple measurement were taken and the greatest of them has been recorded in the database. An image has been attached in the supplementary file labelled as fig S1

Figure 1 and 2 – please label x and y axis and please consider using different colors rather than different subtle shades of blue 

Amendments to figure 2 and 3 ( previously labelled 1 and 2) have been made accordingly, thank you very much 

What kind of ventilator was used? 

GE Datex Ohmeda Avance S5 (Datex-Ohmeda, Ohmeda Drive, PO Box 7550, Madison, Wisconsin, 53707)

Figure 3 is not very clear. The images of the vein look somewhat different in each view and it isn’t clear to me that diameter is really different because it looks like the measurement was not taken at exactly the same spot? 

We do agree, there are some variation in the images. We have noticed such variation in thin patients were chest expansion would greatly alter the ultrasound images. All the measurements would be recorded as the ultrasound is placed while the catheter is also being placed in the subclavian vein. 

I think the main limitations concern how exactly the measurements were done, and possible lack of confidence that they were done the same way every time. It seems like it would be very easy to obtain different and conflicting results depending on how the measurements were done. 

We are extremely grateful for your insightful comment.

We do agree there are some limitation to the methods used to measure the diameter of the subclavian vein however, we have made changes to the methodology while planning this study based on the limitation of previous studies to improve on the quality of evidence. Regardless, this warrants to need for further international collaborations and more prospective studies in favour of bettering patient safety and quality of care

---

## [Decision Letter · Decision Letter 1]

21 Nov 2022

PONE-D-22-12796R1Lung Deflation while Placing a Subclavian Vein Catheter: Our Experience in Minimizing the Risk of Pneumothorax.PLOS ONE

Dear Dr. Rabadi,

Thank you for submitting your manuscript to PLOS ONE. After careful consideration, we feel that it has merit but does not fully meet PLOS ONE’s publication criteria as it currently stands. Therefore, we invite you to submit a revised version of the manuscript that addresses the points raised during the review process. Reviewers' recommendations are below.

We look forward to receiving your revised manuscript.

Kind regards,

Eyüp Serhat Çalık

Academic Editor

PLOS ONE

Additional Editor Comments:

Dear Authors

I reviewed the original and R1 version of your article, the referee suggestions and your answers. First, I see that some of Reviewer 2's recommendations are not fully met. The resolution of the newly added 1st picture is good, but the picture gives the impression that it is not a real intubated patient picture, the sterile covers are very untidy. Please replace it with a higher quality and tidy image. Picture 4 looks the same, please add a new picture based on reviewer 2's suggestions.

In addition, I should say: Add more detailed subheadings in the material and method section. For example, your postprocedural protocol may be a sub-title and under this title you can define complications, the path you follow for diagnosis, etc. You can specify in more detail. You should also make a statistics subtitle and describe the statistical methods you use.

Make sure Reference 1 is written in the correct style.

Your article has been further reviewed by an additional reviewer. You should pay attention to Reviewer 3's recommendations, especially regarding updating references.

Reviewers' comments:

Reviewer's Responses to Questions

**Comments to the Author**

1. If the authors have adequately addressed your comments raised in a previous round of review and you feel that this manuscript is now acceptable for publication, you may indicate that here to bypass the “Comments to the Author” section, enter your conflict of interest statement in the “Confidential to Editor” section, and submit your "Accept" recommendation.

Reviewer #1: All comments have been addressed

Reviewer #3: (No Response)

2. Is the manuscript technically sound, and do the data support the conclusions?

Reviewer #1: Yes

Reviewer #3: Partly

3. Has the statistical analysis been performed appropriately and rigorously? 

Reviewer #1: N/A

Reviewer #3: Yes

4. Have the authors made all data underlying the findings in their manuscript fully available?

Reviewer #1: Yes

Reviewer #3: Yes

5. Is the manuscript presented in an intelligible fashion and written in standard English?

Reviewer #1: Yes

Reviewer #3: Yes

6. Review Comments to the Author

Reviewer #1: An excellewnt manuscript in its field, I have no corrections

I agree with the results

Title: Lung Deflation while Placing a Subclavian Vein Catheter: Our Experience in

Minimizing the Risk of Pneumothorax. Purpose: Lung deflation may reduce the risk of

pneumothorax based on the assumption that the distance between the subclavian vein

and the lung pleura would increase as well as the diameter of the vein. We aim to

provide evidence to support the suggested desideratum of deflation in adults. Methods:

A prospective database was created that included patients who underwent subclavian

vein catheterization for monitoring and therapeutic reasons from January 2014 to

January 2020. Measurements using ultrasonography of the diameter of the subclavian

vein were taken while the patient's breathing was controlled by a ventilator and then

repeated after disconnecting the mechanical ventilation and opening the pressure relief

valve. Results: A total of 123 patients were enrolled, with an average age of 41.9

years. The subclavian vein diameter was measured during controlled breathing with a

mean average of 8.1 ± 0.6mm in males and 7.1 ± 0.5mm in females. The average

increase after lung deflation with the pressure relief valve closed was 8.0± 5.1mm in

males and 13.9 ± 5.4mm in females. An increase was noticed after opening a pressure

valve, and the means were 5.5 ± 2.8mm in males and 5.1 ± 3.3mm in females. The

catheter misplacement rate was 0.8% Conclusion: The benefit of interrupting

mechanical ventilation and lung deflation lies within possibly avoiding pneumothorax as

a complication of subclavian vein catheterization. These findings support the need for

evidence regarding the curtailment of pneumothorax incidence in spontaneously

breathing patients and the suggested increase in first-time punctures and success

rates.

Reviewer #3: PONE-D-22-12796R1

Thank you for providing an updated revision of your original manuscript - I was not a reviewer on the original submission, however I value the opportunity to provide further feedback to all authors.

This manuscript focuses on the use of active lung deflation to prevent pneumothorax during subclavian vein central catheter insertion.

While a novel concept of lung deflation has physiological actions, the benefits are purely focused on mechanically ventilated patient, of which the choice of catheter insertion site may be varied i.e use of IJV is not preferred when patient has a tracheostomy, had recent head/neck surgery or localized trauma in the area, etc..

Please ensure that page numbering is included with your submission, as this makes referencing to areas that require revision easier to pinpoint.

I have also read the previous reviewers comments of R0 and tend to agree somewhat with their points of discussion.

Considering the time-frame this work was performed (6 yrs), the numbers of devices placed was significantly low (n=123), averaging only 20.5 devices/year. This is considered under the minimal number to maintain a high standard of competency based upon published literature.

Was a powered study sample size calculated prior? A sample must be representative of the population, which this only included mechanically ventilated patients with selected inclusion criteria. How does this impact other patients who are critically ill or spontaneously breathing patients who may have a subclavian catheter placed also? Please consider this in your discussions.

Low sample sizes increase the margin of error into the analysis, potentially allowing for insufficient statistical power to answer the primary research question and creating a statistically non-significant result. The authors should clearly address this in the limitations sections of the manuscript.

ABSTRACT - satisfactory.

Please define ‘catheter misplacement’ in the manuscript - it is also mentioned in the abstract as 0.8% and should be described as a’ primary malposition’ as this is determined during the insertion phase. Please see https://doi.org/10.5301/JVA.2011.8381 which describes primary and secondary malpositions/misplacements.

KEYWORDS - please supply 6 keywords using MeSH terms whenever possible to improve searchability. Details can be found here - https://meshb.nlm.nih.gov/

MAIN -

Please try and use third-person perspectives throughout the entire paper when presenting your research e.g. (the authors, this study, these findings, etc.) rather than first-person (we, our, us, etc.) - this makes for a more academically prepared manuscript.

L67 - a femoral catheter tip does not have its tip located in the SVC/IVC as it is not a centrally located device. It would be most likely in the iliac vein at best, depending on total catheter length and patients body habitus. I would avoid stating that femoral catheters are centrally placed UNLESS these devices are over 45-50cm in length, which most traditional CVC’s are not. Correct nomenclature and terminology is important and standardization should be considered when making reference to correct device locations.

Please see the following publications for further details - https://doi.org/10.1177/11297298221126818

L69 - I don’t think its necessary to describe the Seldinger technique - it is already well known. Consider removing.

L73 - there is more recent evidence that supports the use of US guidance, with large number of systematic reviews - consider adding these, as Ref 4 is now a decade old

Suggested readings -

https://doi.org/10.2309/j.java.2019.004.002

https://doi.org/10.1016/j.jemermed.2020.07.039

https://doi.org/10.1007/s00134-019-05564-7

https://doi.org/10.1097/EJA.0000000000001383

https://doi.org/10.1177/0885066619868164

L73 - “Canulating the subclavian vein is part of common medical practice” - firstly, cannulating is spelt incorrectly, please correct.

Secondly, this is often not the case, as many trainees are most commonly taught the IJV approach first, which has been addressed in several publications. The subclavian approach has historically been a ‘high risk approach” however US has improved access to the vessel and actually increased the use of the Axillary vein, which has a clearer approach and visualization when using US. Consider this in your discussion also.

https://doi.org/10.1177/1129729819882602

https://doi.org/10.1007/s00134-019-05651-9

https://doi.org/10.1111/anae.15525

https://doi.org/10.1111/anae.15534

https://doi.org/10.1186/s13613-022-01065-x

https://doi.org/10.1186/s12871-021-01460-0

https://doi.org/10.7759/cureus.23823

https://doi.org/10.1177/11297298211038452

https://doi.org/10.3390/diagnostics12010049

Please create a LIMITATIONS section - while this is addressed at the end of the discussion, it should have its own section.

REFERENCES

17/20 (85%) greater than 4 yrs old. Many of these papers describe practices that are no longer current in today's standards of practice. I would seriously consider updating many of these citations to align with current practices. I have listed a number of quality papers previously that you can consider with your revisions.

FIGURES & TABLES

I would like to see some of the results tabulated for easy reading - currently, they are described in the results section, however the graphs do not do the results justice, making it harder to interpret. Please consider this in your future revisions.

Please ensure all figures have associated text to describe what is portrayed in the image.

Please provide a higher resolution image of Fig’s 2 & 3 - they are difficult to read and they also need a description associated with them. Please correct.

7. PLOS authors have the option to publish the peer review history of their article (what does this mean?). If published, this will include your full peer review and any attached files.

Reviewer #1: **Yes: **paul zarogoulidis

Reviewer #3: No

---

## [Author Response · Author response to Decision Letter 1]

7 Jan 2023

Prof. Emily Chenette

Editor-In-Chief

PLOS ONE

We are very grateful to you for providing us with the opportunity to submit a revised draft of our manuscript titled, “Title: Lung Deflation while Placing a Subclavian Vein Catheter: Our Experience in Minimizing the Risk of Pneumothorax” to Plos One. We are highly appreciative of the time and effort that have been dedicated to providing us with your valuable feedback on the manuscript. We are also thankful to the reviewers for their insightful comments. We have been able to incorporate changes to reflect most of the suggestions provided and have highlighted the changes accordingly in the manuscript. 

Enclosed herewith is a point-by-point response to the reviewers’ comments and concerns. 

Comments Reply

Reviewer #1: 

An excellent manuscript in its field, I have no corrections

I agree with the results Thank you very much. The prospects of improving patient’s quality of care depend on repeated assessments of the measures taken. Therefore, this motivates us to explore unproven methods and validate the necessity to always question way to improve patients safety and outcome of care especially those who critically need it

Reviewer #2 

Considering the time-frame this work was performed (6 yrs), the numbers of devices placed was significantly low (n=123), averaging only 20.5 devices/year. This is considered under the minimal number to maintain a high standard of competency based upon published literature.

Was a powered study sample size calculated prior? A sample must be representative of the population, which this only included mechanically ventilated patients with selected inclusion criteria. How does this impact other patients who are critically ill or spontaneously breathing patients who may have a subclavian catheter placed also? Please consider this in your discussions.

Low sample sizes increase the margin of error into the analysis, potentially allowing for insufficient statistical power to answer the primary research question and creating a statistically non-significant result. The authors should clearly address this in the limitations sections of the manuscript. Thank you very much for such an insightful comment. 

We do agree and there are multiple factors that lead to the low patient numbers. The power indeed calculated during the initial phase of study design. In order for us to achieve 80% power with an alpha rate of 0.05 we needed 141 patients and our initial target was assigned to minimize the rate of type II error to its lowest rate possible. 

However our initial study designed changed due to a significant difference in experience and complication rates per operator which would be a very important factor to eliminate as it is a major limitation to the validity of the anatomical significance this study provides. Having a single sole operator for the study limits the patient recruitment sample available for the study. To add, a significant number of patients have within our recruitment pool have not consented for the procedure and therefore the authors collectively agreed to analyze and publish the data with plans for multicentric international external validations 

 ABSTRACT - satisfactory.

Please define ‘catheter misplacement’ in the manuscript - it is also mentioned in the abstract as 0.8% and should be described as a’ primary malposition’ as this is determined during the insertion phase. Please see https://doi.org/10.5301/JVA.2011.8381 which describes primary and secondary malpositions/misplacements.

 We highly appreciate your input. 

The study protocol defined catheter malposition as a catheter that rests outsides the subclavian vein and the tip is not in its ideal position 

The manuscript text has been edited accordingly as “misplacement” is an English technically vocabulary error. 

KEYWORDS - please supply 6 keywords using MeSH terms whenever possible to improve searchability. Details can be found here - https://meshb.nlm.nih.gov/

 Thank you so much. 

The Keywords have been double check and are infact mesh indexed

Please try and use third-person perspectives throughout the entire paper when presenting your research e.g. (the authors, this study, these findings, etc.) rather than first-person (we, our, us, etc.) - this makes for a more academically prepared manuscript. We are highly thankful for your input and the text has been edited accordingly 

a femoral catheter tip does not have its tip located in the SVC/IVC as it is not a centrally located device. It would be most likely in the iliac vein at best, depending on total catheter length and patients body habitus. I would avoid stating that femoral catheters are centrally placed UNLESS these devices are over 45-50cm in length, which most traditional CVC’s are not. Correct nomenclature and terminology is important and standardization should be considered when making reference to correct device locations.

Please see the following publications for further details - https://doi.org/10.1177/11297298221126818 The manuscript text has been edited accordingly 

I don’t think its necessary to describe the Seldinger technique - it is already well known. Consider removing Thank you so much the manuscript has been edited accordingly 

there is more recent evidence that supports the use of US guidance, with large number of systematic reviews - consider adding these, as Ref 4 is now a decade old

Suggested readings -

https://doi.org/10.2309/j.java.2019.004.002

https://doi.org/10.1016/j.jemermed.2020.07.039

https://doi.org/10.1007/s00134-019-05564-7

https://doi.org/10.1097/EJA.0000000000001383

https://doi.org/10.1177/0885066619868164

Canulating the subclavian vein is part of comm

on medical practice” - firstly, cannulating is spelt incorrectly, please correct.

Secondly, this is often not the case, as many trainees are most commonly taught the IJV approach first, which has been addressed in several publications. The subclavian approach has historically been a ‘high risk approach” however US has improved access to the vessel and actually increased the use of the Axillary vein, which has a clearer approach and visualization when using US. Consider this in your discussion also.

https://doi.org/10.1177/1129729819882602

https://doi.org/10.1007/s00134-019-05651-9

https://doi.org/10.1111/anae.15525

https://doi.org/10.1111/anae.15534

https://doi.org/10.1186/s13613-022-01065-x

https://doi.org/10.1186/s12871-021-01460-0

https://doi.org/10.7759/cureus.23823

https://doi.org/10.1177/11297298211038452

https://doi.org/10.3390/diagnostics12010049

REFERENCES

17/20 (85%) greater than 4 yrs old. Many of these papers describe practices that are no longer current in today's standards of practice. I would seriously consider updating many of these citations to align with current practices. I have listed a number of quality papers previously that you can consider with your revisions. Thank you so much for your input and we highly appreciate your efforts 

We do agree and we have made amendments to the manuscript text however 

There are some articles that quite critical as the evidence regarding anatomical and procedural maneuvers are quite limited. 

The references list has been updated and corrected accordingly as well. Thank 

Please create a LIMITATIONS section - while this is addressed at the end of the discussion, it should have its own section. Thank you so much and this has been done accordingly 

FIGURES & TABLES

I would like to see some of the results tabulated for easy reading - currently, they are described in the results section, however the graphs do not do the results justice, making it harder to interpret. Please consider this in your future revisions.

Please ensure all figures have associated text to describe what is portrayed in the image.

Please provide a higher resolution image of Fig’s 2 & 3 - they are difficult to read and they also need a description associated with them. Please correct. We highly value your input and therefore the figures have been amended and improved as well 

Thank you

---

## [Editor Report · Decision Letter 2]

16 Jan 2023

PONE-D-22-12796R2Lung Deflation while Placing a Subclavian Vein Catheter: Our Experience in Minimizing the Risk of Pneumothorax.PLOS ONE

Dear Dr. Rabadi,

Thank you for submitting your manuscript to PLOS ONE. After careful consideration, we feel that it has merit but does not fully meet PLOS ONE’s publication criteria as it currently stands. Therefore, we invite you to submit a revised version of the manuscript that addresses the points raised during the review process. I have reviewed the R2 version of the manuscript and the responses to the reviewers' criticisms. There are a few minor issues that need to be corrected in the manuscript:

1- Paragraph 1, 2nd and 3rd sentences of the introduction section should be reorganized (lines 65-68), after the change the meaning of the sentences is distorted. 

2- You still have not defined the statistical methods you use in the statistics section (Chi-square test, Fisher's Exact Test, etc.).

3- The resolution of Figure 4 is still very bad, is there any chance to fix it?

We look forward to receiving your revised manuscript.

Kind regards,

Eyüp Serhat Çalık

Academic Editor

PLOS ONE
---

## [Author Response · Author response to Decision Letter 2]

18 Jan 2023

Paragraph 1, 2nd and 3rd sentences of the introduction section should be reorganized (lines 65-68), after the change the meaning of the sentences is distorted. Thank you so much for point the issue out and do acknowledge that it was unclear. 

Alterations to sentences have been 

You still have not defined the statistical methods you use in the statistics section (Chi-square test, Fisher's Exact Test, etc.). We highly appreciate your input and we have described the statistical test used to compare the average means 

 The resolution of Figure 4 is still very bad, is there any chance to fix it? Thank you so much.

We have reproduced the image and focused on minimizing the quality loss while collaging the pictures together. To ensure we provide the best quality we have also PACE ensure the picture meets the journal

---

## [Editor Report · Decision Letter 3]

20 Jan 2023

Lung Deflation while Placing a Subclavian Vein Catheter: Our Experience in Minimizing the Risk of Pneumothorax.

PONE-D-22-12796R3

Dear Dr. Rabadi,

We’re pleased to inform you that your manuscript has been judged scientifically suitable for publication and will be formally accepted for publication once it meets all outstanding technical requirements.

Kind regards,

Eyüp Serhat Çalık

Academic Editor

PLOS ONE
---

## [Editor Report · Acceptance letter]

24 Jan 2023

PONE-D-22-12796R3 

Lung Deflation while Placing a Subclavian Vein Catheter: Our Experience in Minimizing the Risk of Pneumothorax. 

Dear Dr. Rabadi:

I'm pleased to inform you that your manuscript has been deemed suitable for publication in PLOS ONE. Congratulations! Your manuscript is now with our production department. 

Kind regards, 

on behalf of

Dr. Eyüp Serhat Çalık 

Academic Editor

PLOS ONE